# Activation of *Gossypium hirsutum* ACS6 Facilitates Fiber Development by Improving Sucrose Metabolism and Transport

**DOI:** 10.3390/plants12203530

**Published:** 2023-10-11

**Authors:** Chen Geng, Leilei Li, Shuan Han, Mingzhu Jia, Jing Jiang

**Affiliations:** National Key Laboratory of Cotton Bio-Breeding and Integrated Utilization, College of Life Sciences, Henan University, Kaifeng 475004, China; 15188328553@163.com (C.G.); lileilei0408@163.com (L.L.); 10140096@vip.henu.edu.cn (S.H.); 18438700869@163.com (M.J.)

**Keywords:** cotton, *GhACS6.3*, ACC, fiber initiation, yield components, sucrose transport

## Abstract

Cotton fiber yield depends on the density of fiber cell initials that form on the ovule epidermis. Fiber initiation is triggered by MYB-MIXTA-like transcription factors (GhMMLs) and requires a sucrose supply. Ethylene or its precursor ACC (1-aminocyclopropane-1-carboxylic acid) is suggested to affect fiber yield. The *Gossypium hirsutum* (L.) genome contains 35 *ACS* genes (*GhACS*) encoding ACC synthases. Here, we explored the role of a GhACS family member in the regulation of fiber initiation. Expression analyses showed that the *GhACS6.3* gene pair was specifically expressed in the ovules during fiber initiation (3 days before anthesis to 5 days post anthesis, −3 to 5 DPA), especially at −3 DPA, whereas other *GhACS* genes were expressed at very low or undetectable levels. The expression profile of *GhACS6.3* during fiber initial development was confirmed by qRT-PCR analysis. Transgenic lines overexpressing *GhACS6.3* (*GhACS6.3*-OE) showed increased ACC accumulation in ovules, which promoted the formation of fiber initials and fiber yield components. This was accompanied by increased transcript levels of *GhMML3* and increased transcript levels of genes encoding sucrose transporters and sucrose synthase. These findings imply that GhACS6.3 activation is required for fiber initial development. Our results lay the foundation for further research on increasing cotton fiber production.

## 1. Introduction

Cotton is the most important textile fiber crop in the world. One of the determining factors for cotton fiber yield is the frequency of fiber initials that form on the ovule epidermis. However, there is still much to learn about the mechanism of cotton fiber initiation and development. Each cotton fiber is a unicellular seed trichome differentiated from the ovule epidermis [1,2]. There are four distinct and overlapping stages of fiber differentiation and development: differentiation and initiation (3 days before anthesis to 5 days post anthesis, −3 to 5 DPA), rapid elongation or primary cell wall synthesis (5 to 16 DPA), transition from primary wall synthesis to secondary wall synthesis (16 to 20 DPA), secondary cell wall biosynthesis (20 to 40 DPA), and maturation (40 to 50 DPA) [3]. Generally, fiber initial development determines fiber yield by controlling the number of fibers on each ovule. The mechanism underlying this process involves a subclass of MYB transcription factors, among which MYB-MIXTA-like transcription factor 3 (MML3)/GhMML3 regulates fiber initiation [2,4,5,6,7]. A GhMML3-deficient mutant (*N1*) shows a fuzz fiber-less phenotype [8]. Interestingly, GhMML3 also functions as an upstream regulator of other *GhMML* genes, such as *GhMML4* and *GhMML7* [8].

It is well known that cotton ovules use sugars, especially sucrose, to fuel fiber development; that is, the ovule is a sink organ that consumes sucrose. Sucrose produced by the leaves (the source, and the main photosynthetic organ) is transported into ovules for fiber initiation. Studies have shown that the direct product of photosynthesis, glucose, is converted into sucrose in source tissues [9]. It is then loaded into the phloem and transported to sink tissues by two transporters: SWEETs (Sugar Will Eventually be Exported Transporters) and SUTs (Sucrose Transporters) [10,11]. In sink tissues, SuSy (sucrose synthase) converts sucrose into glucose (which then binds uridine diphosphate to form UDPG) and fructose, which provide the building blocks and energy needed for fiber development. Therefore, *SuSy* expression and SuSy activity are directly related to hexose levels in ovules and the fiber initiation rate [12,13,14,15].

Ethylene and its precursor 1-aminocyclopropane-1-carboxylic acid (ACC) have been shown to promote cotton fiber development [15,16,17]. For example, ethylene treatment induces *SuSy* expression, thus promoting fiber initiation [1]. However, the mechanism by which ethylene or ACC regulates SuSy-dependent fiber development is largely unclear. In plants, ethylene is synthesized by two enzymes: ACS (ACC synthase), which converts S-adenosylmethionine (SAM) to ACC [18,19,20], and ACO (ACC oxidase), which oxidizes ACC to generate ethylene [21,22]. There is an increasing body of evidence showing that the activity of cotton ACS or ACO is necessary for fiber development [15]. For example, *G. hirsutum ACS6* (*GhACS6*) is highly expressed in developing ovules [1], and *GhACS6.3* is expressed during fiber initial development [23]. In contrast, a lack of GhACS activity abolishes fiber initial development, as demonstrated by the fact that application of aminoethoxyvinylglycine (AVG), a specific inhibitor of ACS, results in a fiber-free phenotype in cultured ovules [1]. The results of those studies suggest that the activity of cotton ACS family members plays a very important role in fiber initial development. Plant ACS proteins are encoded by multiple genes [24]. For example, the *G. hirsutum* genome contains 35 *ACS* genes [23]. Studies have found that the activity of plant ACS members can be unique, overlapping, and specific [23,24,25,26]. However, it is still unknown which GhACS members are involved in the different stages of fiber initial development, and what their mechanism of action is.

In this study, we analyzed the spatiotemporal expression patterns of *ACS* genes and found that *GhACS6.3* genes are specifically expressed in ovules during fiber initiation (−3 to 5 DPA). Further experiments showed that *GhACS6.3* overexpression increased the ACC content in the ovules, triggered the expression of genes encoding a sucrose metabolism enzyme (GhSuSy) and sugar transporters (GhSWEET and GhSUT), and increased the sucrose and glucose (or UDPG) contents in ovules, which increased the fiber yield components under open field conditions.

## 2. Results

### 2.1. Expression Profiles of GhACS Genes in Ovules and Leaves

To explore the roles of *GhACS* genes during the different stages of fiber initial development, the expression patterns of *GhACS* family members were determined. There are 35 *GhACS* genes in the genome of *G. hirsutum* [23], and transcriptome data are available at the CottonFGD (Cotton Function Genomics Database). Data on the expression of 35 *GhACS* genes in ovules from −3 to 30 DPA and in leaves were downloaded from the CottonFGD, and then the expression profiles of the *GhACS* gene family in various tissues, including the ovule, leaf, root, stem, torus, stamen, pistil, and calycle, were determined and displayed as a heat map.

As shown in the heat map, two *GhACS6.3* genes (*Gh_AACS6.3* in the A genome and *Gh_DACS6.3* in the D genome) were predominantly expressed in the flower (including ovules, stamen, pistil, petal, and calycle) and root (Figure 1A and Appendix A). In ovules, the relative expression levels of this gene pair showed two peaks at −3 DPA and 10 DPA (Figure 1A). In addition to the *GhACS6.3* genes, two *GhACS10.1* genes (*Gh_AACS10.1* in the A genome and *Gh_DACS10.1* in the D genome) were expressed in ovules from −3 to 10 DPA, but their relative expression levels were lower than those of *GhACS6.3*. In leaves, *GhACS10.1* genes were expressed at higher levels than were other *GhACS* genes.

Next, qRT-PCR analyses were conducted to check the expression levels of *GhACS6.3* and *GhACS10.1* in ovules from −3 to 20 DPA. The expression level of *GhACS6.3* reached the maximum at −3 DPA (relative value 0.26), then decreased to a relative value of 0.0036 at 3 DPA (fiber initiation stage), then increased to reach another maximum (relative value 0.19) at 10 DPA (fiber elongation stage), and then rapidly decreased to the minimum (relative value 0) after 10 DPA (Figure 1B). Among the 35 *GhACS* genes, *GhACS6.3* was expressed at significantly higher levels than were *GhACS10.1* genes in ovules from −3 to 0 DPA (Figure 1B). The expression levels of *GhACS6.3* in ovules were 21.06-fold and 38.23-fold higher than that in leaves at −3 DPA and 10 DPA, respectively (Figure 1B,C). *GhACS6.3* and *GhACS10.1* were continuously expressed in the leaves, but the expression level of *GhACS10.1* was higher than that of *GhACS6.3* (Figure 1C). The fact that *GhACS6.3* genes was preferentially expressed in ovules implied that GhACS6 plays an important role for the formation of fiber yield components.

### 2.2. Effects of GhACS6.3 Overexpression on ACC Production in Ovules and Leaves

To explore the relationship between GhACS6.3 and fiber yield components, we examined the fiber yield components in *GhACS6.3*-overexpressing (OE) lines, namely *GhACS6.3*-OE(#1), *GhACS6.3*-OE(#4), and *GhACS6.3*-OE(#5). First, qRT-PCR analyses confirmed the significantly increased mRNA levels of *GhACS6.3* in these lines and showed that *GhACS6.3* was expressed to different degrees in the subtending leaves and the ovules but always at higher levels than that in the control. Among all the OE lines, *GhACS6.3*-OE(#4) had the highest expression levels of *GhACS6.3* (Figure 2A).

Next, we determined the ACC content in the control and the *GhACS6.3*-OE lines. Compared with the control, the *GhACS6.3*-OE lines showed significantly increased ACC production to different degrees in the subtending leaves and the ovules. For example, in *GhACS6.3*-OE(#4) at −3 DPA, the ACC contents in the subtending leaves (117.24 ± 2.95 ng/g) and the ovules (380.97 ± 10.41 ng/g) were significantly higher than those in the control (32.68 ± 1.54 and 211.02 ± 5.28 ng/g, respectively) (Figure 2B). At 0 DPA, the ACC content was higher in the subtending leaves (102.34 ± 9.56 ng/g) and the ovules (251.22 ± 7.50 ng/g) of *GhACS6.3*-OE(#4) than in the control (30.68 ± 1.70 and 104.37 ± 5.69 ng/g, respectively). The transcript level of *GhACS6.3* was directly proportional to the ACC content, indicating that GhACS6.3 increased ACC production in the ovules during fiber initiation. The ACC content was also increased in the leaves of *GhACS6.3*-OE(#4) at −3 and 10 DPA, but the increase was lower than that in the ovules (Figure 2B), implying that GhACS6-dependent ACC production in the ovules may be involved in fiber initial development.

### 2.3. Effects of GhACS6.3-Dependent ACC Production on Fiber Initiation and Yield Components

To explore the relationship between ACC production and fiber yield components, we analyzed the cotton fiber yield components of field-grown plants from 2017. The data from 2021 and 2022 showed that the fiber yield components were clearly increased in the *GhACS6.3*-OE lines compared with the control. For example, in *GhACS6.3*-OE(#4), the number of bolls per plant (43.8 ± 2.6) and the single boll weight (5.6 ± 0.3 g) of *GhACS6.3*-OE(#4) were 18.06% and 24.44% higher, respectively, that their respective values in the control (37.1 ± 2.3 and 4.5 ± 0.3 g, respectively), suggesting that overexpression of *GhACS6.3* gene was beneficial for the development of cotton bolls. 

In addition to the above fiber yield components, the LI (lint index), SI (seed index), LP (lint percentage), and fruit branches of *GhACS6.3*-OE(#4) were higher than those of the control (Table 1).

Then the initial density of fibers on the ovule epidermis was investigated. The fiber cell density (6649 ± 78/mm^2^) was higher in *GhACS6.3*-OE(#4) than in the control (6117 ± 65 /mm^2^) at 0 DPA (Figure 3A,B). The GhACS6.3-dependent ACC production (Figure 2) was positively correlated with the formation of fiber yield components (Table 1).

To prove that ACC facilitates fiber initial development, the effects of ACC treatment on fiber cell initiation from the cultured ovules were tested by adding ACC at different concentrations (0, 0.5, 1.0, 1.5, 2.0, 5.0 μM) into the culture medium. After 48 h, the density of fiber cells on the ovule epidermis was significantly higher in the 0.5, 1.0, and 1.5 μM ACC treatments than in the untreated control (0 μM ACC) (Figure 4). However, high concentrations of ACC (2.0 and 5.0 μM) reduced the density of fiber cells (Figure 4). The size of the cultured ovules did not differ significantly between the ACC treatments and the control (Appendix A). These data further demonstrate that GhACS6.3-dependent ACC production may be a positive regulator of fiber initial development.

Next, the transcript profiles of genes involved in the initiation and development of cotton fibers were determined. The mRNA levels of *GhMML3* in ovules were significantly higher in *GhACS6.3*-OE(#4) than in the control from −1 to 2 DPA. For example, at −1 DPA, the transcript level of *GhMML3* was 2-fold higher in *GhACS6.3*-OE(#4) than in the control (Figure 5A). Accordingly, the transcript levels of *GhMML4* and *GhMML7* in ovules from −1 to 2 DPA were higher in *GhACS6.3*-OE(#4) than in the control (Figure 5B). These findings indicate that ACC produced by GhACS6.3 promoted the expression of *GhMML3*, *GhMML4*, and *GhMML7*, consistent with the increased density of fiber cells (Figure 3) and formation of fiber yield components (Table 1).

### 2.4. Sucrose Content and Transcript Levels of Sucrose Transporter Genes in Leaves and Ovules

Adequate sucrose is necessary for fiber development. The sucrose content in ovules was higher in *GhACS6.3*-OE(#4) than in the control from −1 to 2 DPA (Figure 6). For example, at 0 DPA, the sucrose content in *GhACS6.3*-OE(#4) was 18.47 ± 0.25 mg/g, which was 2.15-fold that in the control (8.60 ± 0.17 mg/g).

Because sucrose is transported from the leaves (source) to the ovules (sink), the effects of *GhACS6.3* overexpression on the transcript levels of genes encoding the sucrose transporters GhSUT and GhSWEET were investigated. Based on RNA-seq data [27], the transcript profiles of 18 *GhSUT* [11] and 21 *GhSWEET* [28] genes in ovules and leaves were determined. As shown in the heat maps, *GhSUT2* was specifically expressed in ovules, *GhSUT1* was predominantly expressed in leaves (Appendix A), *GhSWEET2* specifically expressed in leaves, and *GhSWEET5* was predominantly expressed in ovules (Appendix A).

The results of qRT-PCR analyses showed that, from −1 to 2 DPA, the transcript levels of *GhSUT1* (Figure 7A) and *GhSWEET2* (Figure 7B) in the subtending leaves were higher in *GhACS6.3*-OE(#4) than in the control; and the transcript levels of *GhSUT2* (Figure 7C) and *GhSWEET5* (Figure 7D) in ovules were higher in *GhACS6.3*-OE(#4) than in the control. These results implied that the increased abundance of GhACS6.3 promoted sucrose transport from the leaves to the ovules. 

### 2.5. Involvement of GhACS6.3 in Sucrose Metabolism in Ovules

Ovules are sink organs in which SuSy converts sucrose into UDPG and fructose to support fiber initiation and development. We analyzed the transcript profiles of *GhSuSy* genes that were specifically expressed in the ovule. As shown in the heat map, we determined the transcript profiles of 15 *GhSuSy* [29] in ovules and leaves from −3 DPA to 3 DPA. Of them, *GhSuSy1* and *GhSuSy3* were predominantly expressed in the ovules (Appendix A).

The results of qRT-PCR analyses confirmed that the mRNA levels of *GhSuSy1* and *GhSuSy3* in ovules were higher in *GhACS6.3*-OE(#4) than in the control. For example, at 0 DPA, the relative transcript *GhSuSy1* and *GhSuSy3* in ovules of *GhACS6.3*-OE(#4) were 3.5- and 3.1-times higher than their respective levels in ovules of the control (Figure 8A). Meanwhile, the UDPG (Figure 8B) or fructose (Appendix A) contents showed similar increases in ovules of *GhACS6.3*-OE(#4) from −1 to 2 DPA. These findings suggested that increased activities of GhSuSy2 and GhSuSy3 in *GhACS6.3*-OE plants increased the UDPG and fructose contents to facilitate fiber initial development.

## 3. Discussion

In this study, we detected specific expression of *GhACS6.3* in the ovules during fiber initial development. Our results show that GhACS6.3-dependent ACC production is positively correlated with the fiber yield components and reveal some of the mechanisms underlying increased GhMML3 activity and sucrose transport and utilization.

### 3.1. Specificity of GhACS6.3 Activation during Fiber Initiation

Using data from CottonFGD, we analyzed the transcript profiles of the *GhACS* family in cotton. Among 35 *GhACS* genes [23], a pair of *GhACS6.3* genes was expressed in the flower (including ovules, stamen, pistil, petal, and calycle) and in the root. Notably, from −3 DPA to 0 DPA, the transcript levels of *GhACS6.3* in ovules were higher than those of other *GhACS* genes (Figure 1A,B). Because the period of −3 to 5 DPA corresponds to the fiber initial development stage [7], the transcription of *GhACS6.3* genes during this time may play an important role in fiber initial development. *GhACS6.3*-OE lines showed significantly increased ACC content in ovules from −3 to 10 DPA, and the transcript levels of *GhACS6.3* (Figure 2A) were positively correlated with ACC content (Figure 2B). These results imply that GhACS6.3 catalyzes ACC production in ovules to facilitate fiber initial development.

Unlike *GhACS6.3* (*GH_A12G2512*) specifically involved in cotton fiber initiation, *GhACS6.2* (*GH_D08G1378*; also named *GhACS4* in the website http://www.ncbi.nlm.nih.gov/ (accessed on 6 August 2023) has a gene number of *JF508505*) specifically participated in fiber elongation [30]. Obviously, this specific activation of GhACS6.3 during fiber initiation is reasonable because the activity of plant ACS members is unique, spatiotemporally specific, and overlapping [25,26].

### 3.2. Overexpression of GhACS6.3 Increases Fiber Yield Components

Compared with the control, the transgenic *GhACS6.3*-OE lines showed significantly increased values of fiber yield components, such as the number of bolls per plant, boll weight, LI, SI, LP, and fruit branches under open field conditions (Table 1). These changes were related to the increased density of fiber cell initiation (Figure 3) and ultimately resulted in increased fiber yield components in the *GhACS6.3*-OE lines (Table 1). The LI and SI are the important fiber yield components [31].

The results of this study provide evidence that ACC promotes fiber initiation in cotton. First, both ACC (≤1.5 μM) treatment (Figure 4A,B) and GhACS6.3-produced ACC (Figure 2A,B) increased the density of fiber cell initiation (Figure 3). We explored the molecular and genetic mechanisms by which the density of fiber cell initials was increased in the *GhACS6.3*-OE lines and found that these lines showed increased transcript levels of *GhMML3/4/7* (Figure 5A). GhMML3 is known to control fiber cell initiation [4,8] by regulating the expression of *GhMML4* and *GhMML7* [2,5,32].

### 3.3. Increased Transcript Levels of GhACS6.3 and Increased Sucrose Transport from Leaves to Ovules

During fiber cells’ initiation, cotton ovules need sugar to provide nutrients and material basis for the initiation and development of fiber cells. In addition to an increase of fiber initial development (Figure 3) and yield components (Table 1), the *GhACS6.3*-OE lines showed significantly increased sucrose contents in ovules (Figure 6). These results indicate that GhACS6-produced ACC facilitates sucrose transport from leaves to ovules.

As the most ubiquitous phloem-transported sugar, sucrose is synthesized in source organs (mainly in photosynthetic leaves) and consumed or stored in sink organs [33]. Sucrose is transported between the source and sink organs by transporters such as SUTs and SWEETs [11,34,35]. The cotton ovule is a sink that consumes sucrose, and thus, it requires sucrose to be transported from the leaves. In this study, transcript profiling analyses revealed specific expression of *GhSUT2* in ovules and the predominant expression of *GhSUT1* in the leaves (Appendix A). Consistent with this, a previous study showed that cotton uses a single SUT for phloem loading in mature leaves but uses multiple SUTs for loading into fibers [11]. At the same time, the transcript profiling analyses revealed specific expression of *GhSWEET2* in the leaves and the predominant expression of *GhSWEET5* in ovules (Appendix A), in accordance with the previous finding that the activities of GhSWEET2/5 promote sucrose accumulation in fibers [28]. Our results show that in the *GhACS6.3*-OE lines, the transcript levels of *GhSUT1* and *GhSWEET2* in leaves and those of *GhSUT2* and *GhSWEET5* in ovules were increased from −1 to 2 DPA (Figure 7). These findings imply that in the *GhACS6.3*-OE lines, the cooperation between GhSUT1/2 and GhSWEET2/5 ensures fiber development.

### 3.4. GhACS6.3 Overexpression Affects Sucrose Utilization in Ovules

The activity of SuSy is regarded as a biochemical marker for sink strength, especially in crop species [14]. Sucrose is degraded by SuSy into UPDG and fructose to provide the building blocks and energy necessary for fiber development [12], and this scenario has been proven in upland cotton [29]. In this work, transcript profiling analyses (Appendix A) revealed specific and predominant expression of *GhSuSy1/3* in ovules. Interestingly, compared with the control, the *GhACS6.3*-OE lines showed significantly increased transcript levels of *GhSuSy1/3* (Figure 8A) as well as increased contents of UDPG (Figure 8B) and fructose (Appendix A) in ovules from −1 to 2 DPA. Consistent with this, the transgenic cotton plants with suppressed *GhSuSy* showed reduced fiber initiation [12]. Our results show that an increase in the abundance of GhACS6.3 is beneficial for sucrose metabolism into UDPG and fructose in the young ovules, thus promoting fiber initial development.

## 4. Materials and Methods

### 4.1. Cotton Materials

*G. hirsutum* cv. YZ1 was used as the control. The transgenic *GhACS2/6*-OE lines were created as previously reported [36]. Briefly, the open reading frame of *GhACS6.3* (*GH_A12G2512*) was inserted into the vector pK7WG2 with CaMV 35S promoter and introduced into *Agrobacterium tumefaciens* strain EHA105 and then infected hypocotyl of YZ1. The positive plants were screened on 1/2 MS medium containing 50 μg/mL kanamycin, until obtaining T_3_ lines for research analysis. During cotton plant culturing, seeds were soaked in H_2_O at 25 °C for 24 h and then were germinated on moist cotton wool at 25 °C for 2 days. Cotton young seedings grew in the greenhouse (25 ± 2 °C, 80% relative humidity, 120 μmol m^−2^ s^−1^ light intensity and a 16-h light/8-h dark photoperiod) for two weeks.

According to experimental needs, some seedlings were transplanted into the experimental field in Henan University (Kaifeng, China) with a within-row plant-to-plant distance of 25 to 30 cm. This field experiment has been repeated for many years, and the results are consistent. This manuscript presents data for the two years 2021 and 2022 were averaged over three replications every year with 30 plants per replicate.

### 4.2. Transcriptome Data Analysis

Genome-wide transcriptome data were downloaded from the *G. hirsutum* database (SRA:PRJNA248163) as previously reported [27]. Log_2_(TPM+1) normalization was performed, and the R-4.0.2 language was used to compile the standardized data.

### 4.3. Statistics of Cotton Fiber Yield Components

The methods of statistically determining cotton fiber yield components were used as described in reported literature [31]. The boll weight is the average dry weight of bolls per plant. The lint index (LI) is the lint weight from 100 seeds, and the seed index (SI) is the weight of 100 seeds with fiber. These data of cotton fiber yield components were calculated in 2021 and 2022 employing a complete randomized block design with three replications every year and 30 individuals per replication.

### 4.4. In Vitro Ovule Culturing

The buds were harvested from the −1 DPA ovules (at this point, there was no fibers protruding from the ovule epidermis). Referring to the methods provided in the literature [37], the whole tissue was immediately immersed in 75% ethanol for 5 min with continuous shaking, then rinsed in distilled and deionized water, and soaked again in 0.1% HgCl_2_ solution containing 0.05% Tween-80 for 20 min to sterilize it. Then the shuck of the ovary was removed carefully to avoid damage to the ovules. The ovules were then placed gently on the surface of the medium within each culture plate having 5 mL of nutrient medium. During in vitro culturing, the floating ovules were incubated at 30 ± 1 °C in darkness without shaking.

The ovules were cultured on BT medium with some modifications. The BT medium was prepared at pH 5 according to the literature [37]. According to experimental requirements, the BT media was supplemented with various concentrations of ACC (0, 0.5, 1.0, 1.5, 2.0, or 5.0 μM). The original ACC reagent was purchased from Sigma-Aldrich (Steinheim, Germany). After 48 h of in vitro culturing of the ovules, the fiber initial development on the ovule epidermis was analyzed.

### 4.5. Analysis of the Density of Fiber Cells on the Ovule Epidermis

The number of fiber cells was observed and photographed by using a scanning electron microscope (SEM, FEI Quanta 250, Rock Hill, SC, USA). The number of fiber initial cells was counted from the same middle region of the ovule. The fiber cells in each image were counted by using ImageJ software, and then the density of the fiber initial cells was calculated per 0.01 mm^2^ on the ovules. The experiments were repeated five times with at least 30 ovules per replicate.

### 4.6. RNA Analysis

The ovules and subtending leaves were harvested and immediately immersed in liquid nitrogen and stored at −80 °C. The frozen samples were ground to a fine powder in liquid nitrogen with a mortar and pestle. Total RNA extractions were performed from 100 mg of each macerate of plant tissue using an RNAprep Pure Plant kit (TIANGEN Biotech, Beijing, China). Complementary DNA (cDNA) was synthesized using a Reverse Transcription system (Vazyme, Nanjing, China) and was used as the template for qRT-PCR analyses along with 2× SYBR Green I master mix (Vazyme, Nanjing, China) on a Roche 480 real-time PCR system (Roche, Basle, Sweden). The relative expression levels were calculated based on the comparative ΔC_T_ method [36,38]. The reference gene was cotton *GhUBQ7* (*DQ116441*). The primers used for PCR amplification are listed in Appendix A. 

### 4.7. Measurements of ACC Contents

The ovules and subtending leaves were harvested for ACC extraction. ACC was extracted as described in a previous study [36]. Fresh samples (approximately 100 mg) were harvested and ground to a fine powder in liquid nitrogen, and 1 mL of cold extraction buffer (800 μL of methanol, 190 μL of water, and 10 μL of acetic acid) was added to it. The mixture was homogenized thoroughly by shaking for 16 h at 4 °C in the dark, and it was then centrifuged at 10,000 rpm for 15 min at 4 °C. The supernatant was filtered through the 0.2 μm microfilter and collected. The filtrates were dried using nitrogen gas at room temperature and were then dissolved in 200 μL of methanol. The sample was diluted 100 times and analyzed by using an Applied Biosystems MDS SCIEX 4000 QTRAP liquid chromatography-tandem mass spectrometry system (AB Sciex, Foster City, CA, USA). The ACC standard was used for the quantitative analyses.

### 4.8. Soluble Carbohydrate Analysis

The ovules at −1 to 2 DPA were harvested. Approximately 50 mg samples were ground in liquid nitrogen. The samples were extracted three times with 1 mL of preheated 75% ethanol for 5 min at 80 °C. The aqueous phases of the samples were collected, filtered, and dried under a vacuum at 37 °C and then redissolved in 200 μL distilled water for further analysis. Sucrose, UDP-glucose, and fructose contents were measured and analyzed with an Applied Biosystems MDS SCIEX 4000 QTRAP liquid chromatography-tandem mass spectrometry system (AB Sciex, Foster City, CA, USA). Standard samples (sucrose, UDP-glucose, and fructose) (Sigma-Aldrich, Steinheim, Germany) were used for the quantitative analyses.

### 4.9. Statistical Analysis

At least three biological replicates and three technical replicates were conducted in all of the experiments. Data are presented as means ± SDs. Asterisks indicate statistically significant differences, as determined by Student’s *t*-tests (*, *p* < 0.05 indicate significant differences, **, *p* < 0.01 indicate extremely significant differences).

## 5. Conclusions

During fiber initial development, GhACS6.3-dependent ACC accumulation in the ovule promotes GhMML3-triggered fiber initiation and improves sucrose transport to the ovules and metabolism in the ovules and thus increases fiber yield components. This study provides new ideas for improving cotton fiber yield components.

## Figures and Tables

**Figure 1 plants-12-03530-f001:**
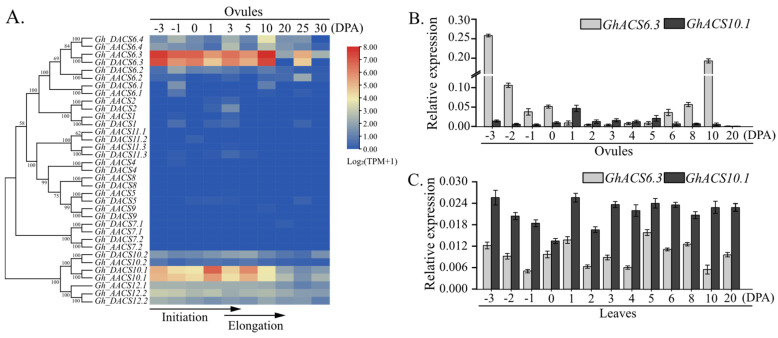
Expression profiles of *GhACS* genes in ovules and leaves during fiber development. (**A**) Phylogenetic analysis of 35 *GhACSs*. Neighbor-joining tree was constructed using MEGA 7.0 software. Heat-map was generated from RNA-seq data of *GhACSs* in ovules at −3, −1, 0, 1, 3, 5, 10, 20, 25, and 30 DPA. Colored bar represents gene transcript level (log_2_ (TPM+1)) in the sample. (**B**,**C**). qRT-PCR analysis of transcript levels of *GhACS6.3* and *GhACS10.1* in ovules and leaves at −3 to 6, 8, 10, and 20 DPA, respectively, normalized against *GhUBQ7* as the internal control. Experiments were repeated three times with similar results. Values are means ± SDs.

**Figure 2 plants-12-03530-f002:**
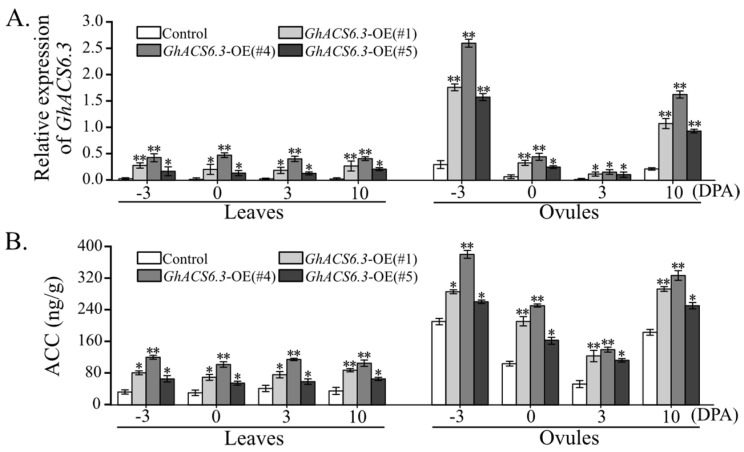
Effect of *GhACS6.3* overexpression on ACC production in ovules. (**A**). qRT-PCR analysis of transcript levels of *GhACS6.3* in leaves and ovules at −3, 0, 3, 10 DPA. (**B**). HPLC analysis of ACC levels in leaves and ovules at −3, 0, 3, 10 DPA, normalized against *GhUBQ7* as the internal control. Experiments were repeated three times with similar results. Values are means ± SDs (Student’s *t*-test; * *p* < 0.05, and ** *p* < 0.01).

**Figure 3 plants-12-03530-f003:**
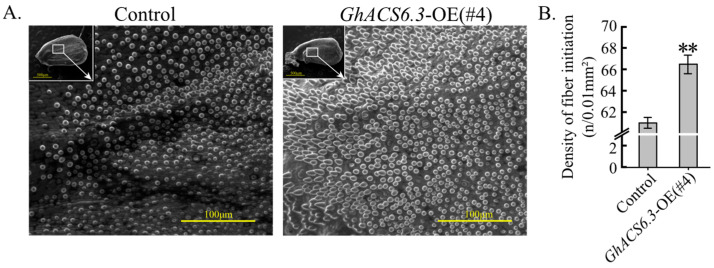
Fiber initial density of *GhACS6.3*-OE lines grown under open field conditions. (**A**) Images of ovule epidermis at 0 DPA. Scale bar = 500 μm for whole ovules and 100 μm in higher-magnification images. (**B**) Statistical analysis of fiber cell density on ovule epidermis of control and *GhACS6.3*-OE(#4) lines. Experiments were repeated five times with similar results, with at least 30 ovules per replicate. Values are means ± SDs (Student’s *t*-test; ** *p* < 0.01).

**Figure 4 plants-12-03530-f004:**
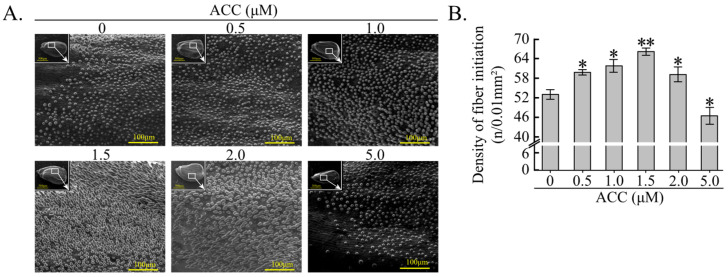
Density of fiber initials under ACC treatment. (**A**) Images of fiber cells on ovules subjected to 0, 0.5, 1.0, 1.5, 2.0, and 5.0 μM ACC treatment for 48 h. (**B**) Statistical analysis of fiber cell density on ovule epidermis under 0, 0.5, 1.0, 1.5, 2.0, and 5.0 μM ACC treatment for 48 h. Experiments were repeated five times with similar results, with at least 30 ovules per replicate. Values are means ± SDs (Student’s *t*-test; * *p* < 0.05, and ** *p* < 0.01).

**Figure 5 plants-12-03530-f005:**
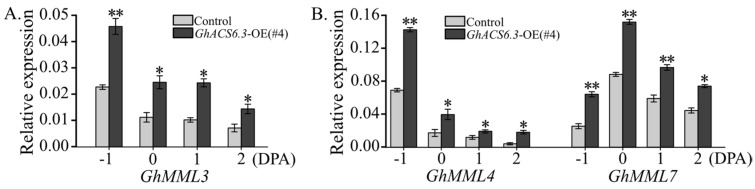
Transcript profiles of *GhMML3/4/7* in ovules of *GhACS6.3*-OE lines and wild-type control. (**A**,**B**) qRT-PCR analysis of transcript levels of *GhMML3*, *GhMML4*, and *GhMML7* in ovules at −1 to 2 DPA, normalized against *GhUBQ7* as the internal control. Experiments were repeated three times with similar results. Values are means ± SDs (Student’s *t*-test; * *p* < 0.05, and ** *p* < 0.01).

**Figure 6 plants-12-03530-f006:**
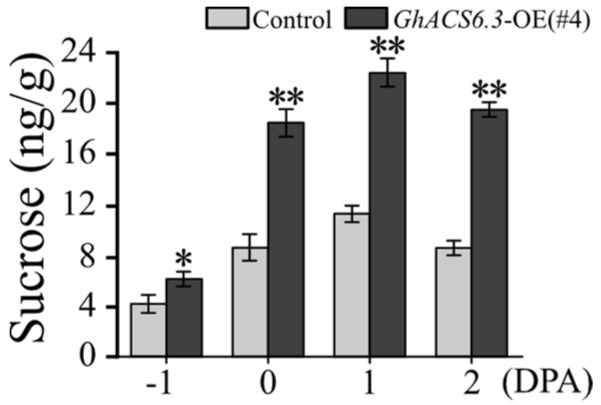
Sugar content in ovules of *GhACS6.3*-OE lines. HPLC analysis of sucrose in ovules at −1 to 2 DPA. Experiments were repeated three times with similar results. Values are means ± SDs (Student’s *t*-test; * *p* < 0.05, and ** *p* < 0.01).

**Figure 7 plants-12-03530-f007:**
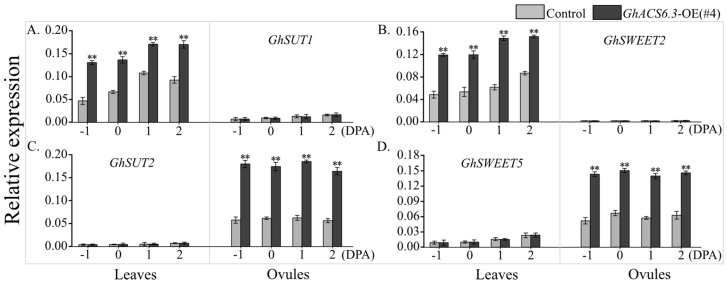
Transcript profiles of *GhSUTs* and *GhSWEETs* in ovules and leaves. (**A**–**D**) qRT-PCR analysis of transcript levels of *GhSUT1*, *GhSUT2*, *GhSWEET2*, and *GhSWEET5* in leaves and ovules at −1 to 2 DPA, normalized against *GhUBQ7* as the internal control. Experiments were repeated three times with similar results. Values are means ± SDs (Student’s *t*-test; ** *p* < 0.01).

**Figure 8 plants-12-03530-f008:**
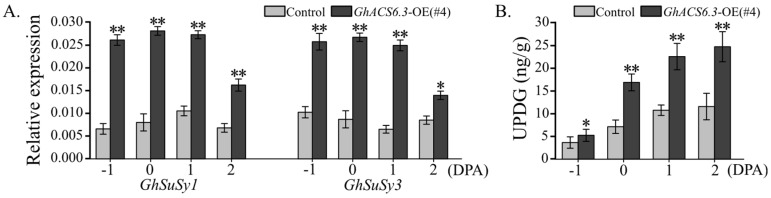
Transcript levels of *GhSuSys* in ovules of *GhACS6.3*-OE lines. (**A**) qRT-PCR analysis of mRNA levels of *GhSuSy1* and *GhSuSy3* genes in ovules at −1 to 2 DPA, normalized against *GhUBQ7* as the internal control. Experiments were repeated three times with similar results. Values are means ± SDs (Student’s *t*-test; * *p* < 0.05, and ** *p* < 0.01). (**B**) HPLC analysis of UPDG in ovules at −1 to 2 DPA. Experiments were repeated three times with similar results. Values are means ± SDs (Student’s *t*-test; * *p* < 0.05, and ** *p* < 0.01).

**Table 1 plants-12-03530-t001:** Yield components of *GhACS6.3*-OE lines under open field conditions.

		2021	2022	Average
Bolls per plant	Control	35.5 ± 1.7	38.7 ± 1.6	37.1 ± 2.3
	*GhACS6.3*-OE(#4)	45.4 ± 2.4 **	42.2 ± 1.7 *	43.8 ± 2.6 **
Boll weight (g)	Control	4.5 ± 0.2	4.6 ± 0.3	4.5 ± 0.3
	*GhACS6.3*-OE(#4)	5.4 ± 0.2 *	5.8 ± 0.2 **	5.6 ± 0.3 *
LI (g)	Control	7.6 ± 0.2	7.3 ± 0.2	7.5 ± 0.2
	*GhACS6.3*-OE(#4)	8.3 ± 0.3 *	7.9 ± 0.3 *	8.1 ± 0.3 *
SI (g)	Control	8.8 ± 0.3	9.3 ± 0.3	9.0 ± 0.4
	*GhACS6.3*-OE(#4)	9.4 ± 0.2 **	9.7 ± 0.2 *	9.5 ± 0.3 *
Lint percentage (%)	Control	46.3 ± 1.1	44.1 ± 1.0	45.2 ± 1.5
	*GhACS6.3*-OE(#4)	46.8 ± 1.0 *	45.1 ± 1.1 **	46.0 ± 1.4 **
Plant height (cm)	Control	119.8 ± 9.9	113.7 ± 7.7	116.8 ± 9.4
	*GhACS6.3*-OE(#4)	122.5 ± 11.5	114.8 ± 8.8	118.7 ± 10.9
Fruit branches	Control	17.5 ± 1.5	17.4 ± 1.7	17.5 ± 1.6
	*GhACS6.3*-OE(#4)	17.9 ± 1.5	18.2 ± 1.2 *	18.0 ± 1.3 *

Bolls per plant = the total number of mature bolls, young bolls, and batting bolls, Boll weight = the average dry weight of all batting bolls per plant, LI (g) = the lint weight of 100 unginned seeds, SI (g) = the weight of 100 seeds, Lint percentage (%) = lint weight (g)/unginned seeds weight (g) × 100. These cotton fiber yield components showed in 2021 and 2022 with a complete randomized block design with three replications every year, and 30 individuals per replication were employed. Asterisks indicate statistically significant differences between the control and *GhACS6.3*-OE(#4) as determined by Student’s *t*-test (* *p* < 0.05, ** *p* < 0.01). Error bars represent ± SD.

## Data Availability

All data relevant to this study are presented in figures and Appendix A and can be found in online repositories. Sequence data for the genes described in this study can be found in the databases of *G. hirsutum* (https://www.cottongen.org/ (accessed on 6 August 2023)) under the accession numbers: *GhACS6* (*Gh_A12G2673*; *GhACS6_A*). The control gene can be found in the NCBI database (https://www.ncbi.nlm.nih.gov/genbank/ (accessed on 18 August 2023)) under the accession number: *GhUBQ7* (*DQ116441*).

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
