# Peer review of "Activation of Gossypium hirsutum ACS6 Facilitates Fiber Development by Improving Sucrose Metabolism and Transport"

_plants, 2023, doi:10.3390/plants12203530_

Round 1
Reviewer 1 Report
The study generated transgenic cotton plants overexpressing one of the genes encoding ACC synthase, GhACS6.3. As a result, the content of ACC, the precursor of ethylene biosynthesis, increased in the cotton plants overexpressing GhACS6.3. The transgenics also had improved yield components, including boll number per plant, boll weight, lint index and seed index. The authors inferred that the improved yield components could be a result of enhanced expression of the key gene involved in fiber initiation and the activity of sugar transportation and metabolism.
While the role of ACS genes in cotton fiber initiation and development has been reported previously, based on gene expression data, this study selected the members of the gene families involved in sucrose transportation and metabolism and showed high expression during fiber development, and investigated their responses to the enhanced expression of GhACS6.3. The results help us understand the gene networks associated with fiber initiation and development, so I would suggest acceptance of the manuscript after the authors addressing the following issues:
1. The phenotypic data presented in the manuscript were boll number/plant, boll weight, lint index and seed index. They are yield components but not yield per se, so using of “fiber yield” was not justified and need to be changed throughout the manuscript.
2. It’s surprising to know that overexpressing GhACS6.3 significantly increased all the four yield components investigated. Increased boll number per plant could be a result of enhanced vegetative growth of the transgenic plants, i.e., the transgenics might have a larger statue than the control, consequently having more fruit branches and more fruit nodes. The authors are suggested to provide the information of traits like plant height, the number of main stem nodes (true leaves), total number of fruit nodes per plant, and lint percentage. Also, did the authors observe negative correlation between lint index and seed index in the transgenic plants?
3. According to the results reported by Song et al. (2023) [Plant Biotechnology Journal, doi: 10.1111/pbi.14138], it seems that the major ACS gene regulating fiber initiation and development is GhACS4. Is the GhACS4 reported by Song et al. (2023) equivalent to GhACS6.3 reported in this study or they are different (if so, how can the authors reconcile their results with that of Song’s?
4. The use of present sense and past sense was quite confusing throughout the manuscript. The entire manuscript needs to be revised to correct the grammatical errors.
5. The first two subsections of the Discussion were basically repeating the results and need revision to avoid repetition.
see above
Reviewer 2 Report
Summary/Goals: Authors seek to explore the role of ACC synthase in fiber initiation using transcription analysis, qRT-PCR, and overexpression lines of GhACS6.3. Findings support that overexpression of ACS leads to ACC accumulation and promotes increased fiber initiation, leading to higher fiber yield. Authors utilized previously published transcriptome data from 2022.
Overall, the manuscript accomplished the goals set by the authors and provides a convincing case for ACS having an important role in cotton fiber initiation. The manuscript advances knowledge in the field of cotton fiber development
Introduction – why is this research important? Why use cotton as a study system?
Conclusions/Discussion - Are there GhACS6.3 knockouts/knockdowns available? Could they be generated? Does GhACS6.3 have similar expression patterns in G. barbadense?
[28-32] It has been shown that elongation can continue past 20DPA in elite hirsutum lines
[32-33] It would be worthwhile to consider the proportion of fiber initials that form fuzz fibers as opposed to lint fibers
[42-48] Section could be replaced by an explanation of SuSy function and how it impacts fiber initial differentiation
[129-132] Please explain how the fact that the concentration was lower in leaves is connected to fiber initiation
[214-218] Is it known what the functional/expression differences are between these genes in Arabidopsis or other plants?
[259-261] Which genes have overlapping functions with GhACS6.3, if any?
The manuscript, while legible, contains numerous grammatical errors that make reading somewhat challenging and distracting. I would recommend the manuscript be revised by someone proficient in English writing.
Round 2
Reviewer 1 Report
For the revisions that have been done in the revised manuscript, they need to be clearly present in the response letter, so the reviewer does not necessarily to go through the whole manuscript to find out what have been changed.
“Fiber yield” is still being used. “Yield index” is not used by the cotton community, instead, “yield component” should be used.
Suppl. Table 2 can be merged with Table 1.
Song’s work on GhACS4 and the function of GhACS4 in fibre elongation should be introduced in the Introduction and discussed in the context of the function od GhACS6.3 presented in this study. Current, GhACS4 was not mentioned at all.
There are still grammatical errors.
Author Response
Q: 1. “Fiber yield” is still being used. “Yield index” is not used by the cotton community, instead, “yield component” should be used.
Response:
We have replaced the yield index with the yield component in the revised manuscript.
Q: 2. Suppl. Table 2 can be merged with Table 1.
Response:
We agree with this suggestion and have merged Supplementary Table 2 with Table 1, and thus deleted Supplementary Table 2.
Q: 3. Song’s work on GhACS4 and the function of GhACS4 in fiber elongation should be introduced in the Introduction and discussed in the context of the function of GhACS6.3 presented in this study. Current, GhACS4 was not mentioned at all.
Response:
In the Discussion section of the revised manuscript, we discussed the specific role of GhACS6.3 in fiber initiation and GhACS4 (GhACS6.2) in fiber elongation, and added this literature to the Reference section (highlighted in yellow)
Q: 4. Comments on the Quality of English Language. There are still grammatical errors.
Response:
We have tried our best to checked the whole manuscript, and have also involved native English speakers for language corrections.
